# The Safety and Immunogenicity of the BNT162b2 mRNA COVID-19 Vaccine in Japanese Patients after Allogeneic Stem Cell Transplantation

**DOI:** 10.3390/vaccines10020158

**Published:** 2022-01-21

**Authors:** Marika Watanabe, Kimikazu Yakushijin, Yohei Funakoshi, Goh Ohji, Wataru Hojo, Hironori Sakai, Miki Saeki, Yuri Hirakawa, Sakuya Matsumoto, Rina Sakai, Shigeki Nagao, Akihito Kitao, Yoshiharu Miyata, Taiji Koyama, Yasuyuki Saito, Shinichiro Kawamoto, Mitsuhiro Ito, Tohru Murayama, Hiroshi Matsuoka, Hironobu Minami

**Affiliations:** 1Division of Medical Oncology/Hematology, Department of Medicine, Kobe University Graduate School of Medicine, Kobe 650-0017, Japan; mwat624@med.kobe-u.ac.jp (M.W.); yohei@med.kobe-u.ac.jp (Y.F.); msaeki20@med.kobe-u.ac.jp (M.S.); hirakawa@med.kobe-u.ac.jp (Y.H.); samatsu@med.kobe-u.ac.jp (S.M.); rsakai@med.kobe-u.ac.jp (R.S.); shigeki.nagao@gmail.com (S.N.); akitao@med.kobe-u.ac.jp (A.K.); hameyama@med.kobe-u.ac.jp (T.K.); skawamo@med.kobe-u.ac.jp (S.K.); itomi@med.kobe-u.ac.jp (M.I.); tmurayam@hp.pref.hyogo.jp (T.M.); hminami@med.kobe-u.ac.jp (H.M.); 2Division of Infectious Disease Therapeutics, Department of Microbiology and Infectious Diseases, Kobe University Graduate School of Medicine, Kobe 650-0017, Japan; ohji@med.kobe-u.ac.jp; 3R&D, Cellspect Co., Ltd., Morioka 020-0857, Japan; whoujou@cellspect.com (W.H.); hsakai@cellspect.com (H.S.); 4BioResource Center, Kobe University Hospital, Kobe 650-0047, Japan; yhmiyata@med.kobe-u.ac.jp (Y.M.); matsuoh@med.kobe-u.ac.jp (H.M.); 5Division of Molecular and Cellular Signaling, Kobe University Graduate School of Medicine, Kobe 650-0017, Japan; ysaito@med.kobe-u.ac.jp; 6Laboratory of Hematology, Division of Medical Biophysics, Kobe University Graduate School of Health Sciences, Kobe 654-0142, Japan; 7Hematology Division, Hyogo Cancer Center, Akashi 673-0021, Japan; 8Cancer Center, Kobe University Hospital, Kobe 650-0017, Japan

**Keywords:** hematopoietic stem cell transplantation, BNT162b2, COVID-19, vaccination

## Abstract

Patients who have undergone hematopoietic stem cell transplantation (HSCT) for hematological disease experience high mortality when infected by coronavirus disease 2019 (COVID-19). However, the safety and efficacy of the COVID-19 vaccine in HSCT patients remain to be investigated. We prospectively evaluated the safety and immunogenicity of the BNT162b2 mRNA COVID-19 vaccine (Pfizer BioNTech) in 25 Japanese allogeneic HSCT patients in comparison with 19 healthy volunteers. While anti-S1 antibody titers in almost all healthy volunteers after the second dose were higher than the cut-off value reported previously, levels in HSCT patients after the second dose were diverse. Nineteen patients (76%) had seroconversion of anti-S1 IgG. The median optical density of antibody levels in HSCT patients with low IgG levels (<600 mg/dL), steroid treatment, or low lymphocytes (<1000/μL) was significantly lower than that in the other HSCT patients. There were no serious adverse events (>Grade 3) and no new development or exacerbation of graft-versus-host disease after vaccination. We concluded that the BNT162b2 mRNA vaccine is safe and effective in Japanese allogeneic HSCT patients.

## 1. Introduction

Patients with previous hematopoietic stem cell transplantation (HSCT) who are affected by coronavirus disease 2019 (COVID-19) are reported to have a high mortality rate of around 30% [1]. HSCT patients are known to be highly immunocompromised due to past chemotherapy or immunosuppressive agents for graft-versus-host disease (GVHD). Prevention of COVID-19 is therefore critically important in these patients, while some treatments, including immunotherapy, are considered to be promising strategies against COVID-19 [2,3].

The BNT162b2 mRNA COVID-19 vaccine (Pfizer BioNTech) was developed in 2020 and induces robust antibody and T cell responses for SARS-CoV-2 [4]. The phase 3 trial of this vaccine reported 95% efficacy in preventing symptomatic COVID-19 [5]. However, the safety and efficacy of this vaccine for HSCT patients remain to be investigated because these patients were excluded from the initial registration trials. Some studies have reported the safety and efficacy of SARS-CoV-2 vaccines for HSCT patients [6,7,8]. Efficacy and safety need to be determined with regard to allogeneic transplantation and ethnic differences, as is also the case with GVHD [9]. A pivotal study showed that Asian people tended to have lower vaccine efficacy than other ethnicities, albeit the difference is not significant [10]. Further, racial variation in the major histocompatibility complex molecules may affect the results of vaccination [11]. With regard to safety, it has been reported that BNT162b2 might induce controllable GVHD or cytopenia with mixed chimerism between donor and recipient [12], but the safety and immunogenicity of the COVID-19 vaccines for Japanese patients who have undergone HSCT are unclear.

Here, we prospectively investigated the safety and immunogenicity of BNT162b2 in Japanese patients who had undergone HSCT.

## 2. Patients and Methods

### 2.1. Study Design

Consecutive patients who had undergone HSCT at Kobe University Hospital were enrolled, together with healthy volunteer controls. All patients who were recruited at least three months after transplantation and who gave written informed consent to participate in the study between March 2021 and August 2021 were included. All participants were vaccinated with 2 doses of BNT162b2 with a 3-week interval. Peripheral blood samples were collected at pre-vaccination (within 14 days prior to the first dose), within 7 days prior to the second dose and 14 days (+/− 7 days) after the second dose of BNT162b2. Exclusion criteria included a documented COVID-19 infection within 14 days of the second dose. Vaccine-related adverse events were evaluated for 28 days after vaccination by Common Terminology Criteria for Adverse Events 5.0 except for fever, which we defined as Grade 1 (37.5–37.9 °C), Grade 2 (38.0–38.9 °C), Grade 3 (39.0–39.9 °C) and Grade 4 (≥40.0 °C), in the armpit. This study was approved by the Kobe University Hospital Ethics Committee (No. B2056714) and was conducted in accordance with the Declaration of Helsinki. All participants provided written informed consent for this study.

### 2.2. Sample Collection and Measurement of Antibody Titers

Serum samples were obtained by centrifugation of blood samples at 1000× *g* for 10 min at room temperature and immediately transferred to a freezer kept at −80 °C.

Antibody titers against S1 were measured using the QuaResearch COVID-19 Human IgM IgG ELISA kit (Spike Protein-S1) (Cellspect, Inc., RCOEL961S1, Iwate, Japan). This kit is based on the indirect ELISA method and comes with different immobilized antigenic proteins. The plate of the ELISA kit (Spike Protein-S1) is immobilized with a recombinant spike protein (S1, 251-660AA) of SARS-CoV-2 expressed in *Escherichia coli*. Serum samples were diluted 1:200 in 1% BSA/PBST for RCOEL961S1. The plates were read at 450 nm with an SH-1200 plate reader (Corona Electric Co. Ltd. Hitachinaka, Japan) in accordance with the manufacturer’s protocol. Based on a previous report [13], the optimal optical density (O.D.) cut-off value of the anti-S1 IgG antibody for seroconversion was determined to be 0.26.

### 2.3. Statistical Analysis

Categorical and continuous variables were compared using Fisher’s exact test and the Mann–Whitney *U* test, respectively. Pearson’s test was used to evaluate correlations between lymphocyte count and IgG level. All statistical tests were two-sided and were performed using STATA (version 17.0; Stata Corp, TX, USA.) and EZR (Saitama Medical Center, Jichi Medical University, Saitama, Japan), a graphical user interface for R (The R Foundation for Statistical Computing, Vienna, Austria) [14] with *p* < 0.05 as the level of significance.

## 3. Results

### 3.1. Patient Characteristics

Twenty-five patients who underwent HSCT and nineteen healthy volunteers were included in this study. Patient characteristics are shown in Table 1.

The median age of the HSCT patients was 55 years (range, 23–71 years) at the time of the first dose and 11 patients were female. The median duration between HSCT and vaccination was 1605 days (range, 163–4126 days). Only one patient who received rituximab therapy was included, and the last administration was over five years ago. Seven patients were receiving immunosuppressants at the time of the first vaccination. Two of these seven patients were taking tacrolimus alone and one was receiving tacrolimus plus concomitant steroids. Three patients took steroids alone and the other received both cyclosporine and steroid for nephrotic syndrome. Hematological disease status was stable in all patients, and none had developed COVID-19 prior to the first dose. The median level of serum IgG was 1087 mg/dL (range, 181–2169), and the median absolute lymphocyte count was 2320/μL (range, 309–4998). Regarding the 19 healthy volunteers used as a control (female, *n* = 7), the median age was 74 years (range, 39–82), and none had developed COVID-19 infection prior to the study.

### 3.2. Serological Outcomes

Anti-S1 IgG antibody titers in healthy volunteers were all significantly higher after the second dose than at pre-vaccination, and all except one participant experienced seroconversion. In contrast, titers in HSCT patients after the second dose were diverse. The median O.D. of anti-S1 IgG antibody titers after the second dose in HSCT patients and the healthy volunteer group were 0.540 (range, 0.016–1.991) and 0.687 (range, 0.259–1.498), respectively (*p* = 0.41) (Figure 1). Nineteen patients (76%) had higher anti-S1 antibody titers than the threshold (0.26) for seroconversion.

Results for the HSCT patients by subgroup are shown in Figure 2. The median O.D. of antibody levels in patients with a high or low IgG level (≥ or <600 mg/dL), with or without steroid treatment, and with high or low lymphocytes (≥ or <1000/μL) were 0.739 (range, 0.037–1.991) vs. 0.097 (range, 0.016–0.417) (*p* = 0.01), 0.7655 (range, 0.037–1.991) vs. 0.121 (range, 0.016–0.417) (*p* = 0.01), and 0.792 (range, 0.037–1.991) vs. 0.2625 (range, 0.016–0.509) (*p* = 0.01), respectively. However, we found a moderate correlation between lymphocyte count and IgG titer (r = 0.651). There was no significant difference in anti-S1 IgG antibody titer between the group taking calcineurin inhibitors and those not taking them (*p* = 0.45). Multivariate analyses could not be performed due to the small number of cases.

The scatter plot of anti-S1 IgG titers by days from HSCT to first immunization is shown in Figure 3. Four patients received vaccination within one year after HSCT, two of whom had higher anti-S1 antibody titers than 0.26. Of the 21 patients who received vaccination more than one year after HSCT, 17 (81%) had anti-S1 antibody titers higher than 0.26 (*p* = 0.23).

### 3.3. Safety

Data on vaccine-related adverse events are shown in Table 2. No serious adverse events (>Grade 3) were reported in any participant. The most frequent adverse event was pain at the injection site. Adverse events after the second dose were slightly worse than those after the first dose in our HSCT patients. In this study, no newly diagnosed GVHD or GVHD exacerbation was encountered after vaccination.

## 4. Discussion

We prospectively investigated the safety and immunogenicity of BNT162b2 in Japanese HSCT patients. Six patients had antibody levels lower than 0.26. However, 76% of patients obtained sufficient immunogenicity. Recent studies have reported that seroconversion after two doses of vaccination reaches 75–86% in allogeneic HSCT patients [7,12,15,16,17,18]; our present results are consistent with this level. In one previous report, the main factor influencing vaccination response was the time elapsed from HSCT, with lower responses occurring within one year of HSCT [15]. In our present subjects, one of the four patients vaccinated within one year from transplantation had an apparently low antibody titer, and no association with antibody titer was seen for the interval between transplantation and first dose, possibly due to the small number of patients who underwent allogeneic stem cell transplantation within 1 year before vaccination. Immunosuppressive therapy [16,17,18] and chronic GVHD [16,18] at vaccination are reported to be associated with a low response to SARS-CoV-2 vaccination. In our cohort, only three patients had developed chronic GVHD and were taking steroids. Additionally, one patient received steroids and cyclosporin against nephrotic syndrome. In our study, calcineurin inhibitors were not associated with a low response to vaccination, whereas a low total lymphocyte count, steroid use and low total IgG level were associated with poor response to vaccination; however, we were unable to perform multivariate analyses because of the small number of the cases. The actual impact of calcineurin inhibitors on vaccine response is unclear.

In our cohort, all adverse events were mild, and pain at the injection site was the most common event, as previously reported [16,17,18]. A recent study reported that approximately 10% of patients developed cytopenia after each dose, which resolved within two weeks [12]. As we did not routinely perform blood examination one week after each dose, we might not have been able to detect cytopenia as an adverse event; nevertheless, no patients experienced infection or bleeding events soon after vaccination. In terms of GVHD, while exacerbation or development of GVHD was observed in some previous reports [12,16,18], only a few patients (4.5%) experienced exacerbation of GVHD, all cases of which developed within the first week of injection in one report [12]. In that report, all cases of GVHD exacerbation were easy to control. In our study, no patient developed GVHD or experienced an exacerbation of GVHD, which suggests that BNT162b2 could also be safely administered to Japanese HSCT patients. Interestingly, the possibility that a third vaccination may be effective without reactivation of GVHD was recently reported [19]. In our cohort, there were six patients who did not obtain sufficient immunogenicity; a third vaccination should be considered in patients who seem unlikely to acquire sufficient immunity with two doses of vaccination, albeit further investigation of this possibility is required.

Several limitations of our study warrant mention. First, the cohort was too small to allow multivariate analyses. Second, as the disease status in our cohort was stable, safety and immunogenicity in patients with relapse or receiving high-dose steroids for acute or chronic GVHD are uncertain. Third, the precise titer borderline providing clinical protection for COVID-19 remains unclear. Fourth, the observation period might have been too short to evaluate late-onset adverse events, including GVHD. However, we believe that many Japanese HSCT patients would have gained sufficient immunity, as did the healthy volunteers.

## 5. Conclusions

To our knowledge, this study is the first prospective observational study on the safety and immunogenicity of BNT162b2 in Japanese patients who received allogeneic HSCT. Most HSCT patients obtained certain immunogenicity without severe adverse events, whereas patients taking steroids or with low lymphocyte counts did not yield sufficient antibody titers. Further investigation to confirm our findings is needed, and long-term monitoring of antibody levels and late-onset adverse events, including GVHD, is warranted.

## Figures and Tables

**Figure 1 vaccines-10-00158-f001:**
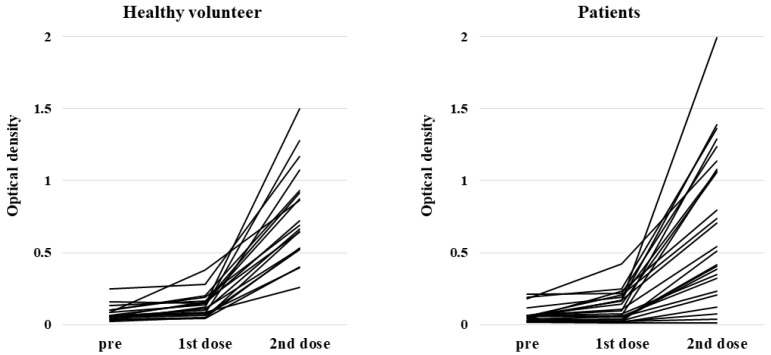
Anti-S1 antibody response at pre-vaccination (within 14 days prior to the first dose), within 7 days prior to the second dose and 14 days (+/− 7 days) after the second dose of BNT162b2 in healthy volunteers and patients with hematopoietic stem cell transplantation.

**Figure 2 vaccines-10-00158-f002:**
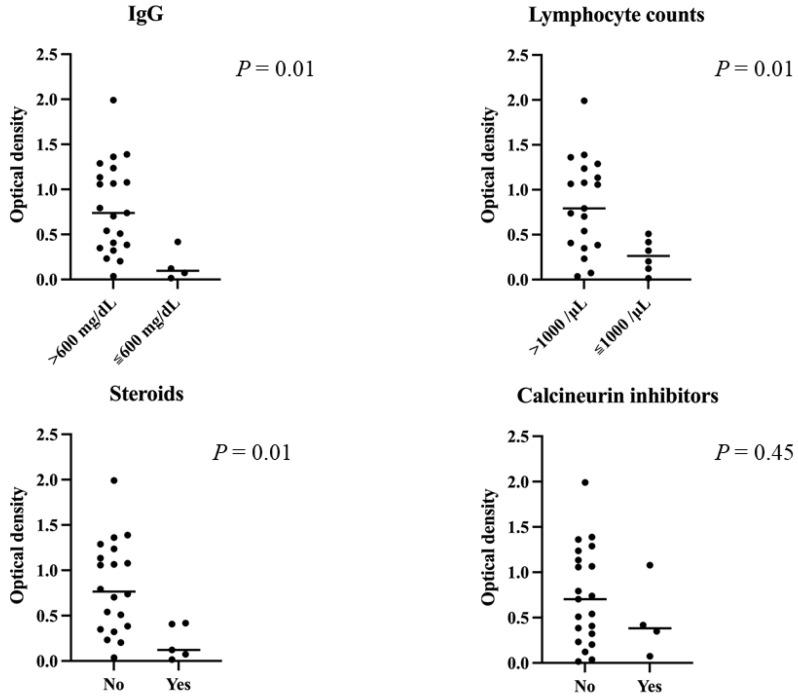
Anti-S1 titers after the second dose in each subgroup of transplant patients. Median optical density of antibody levels in patients with low IgG levels (<600 mg/dL), steroid treatment and low lymphocytes (<1000/μL) was significantly lower than in the other patients. There was no significant difference in S1-antibody titers between the group taking calcineurin inhibitors and the group not taking them (*p* = 0.45).

**Figure 3 vaccines-10-00158-f003:**
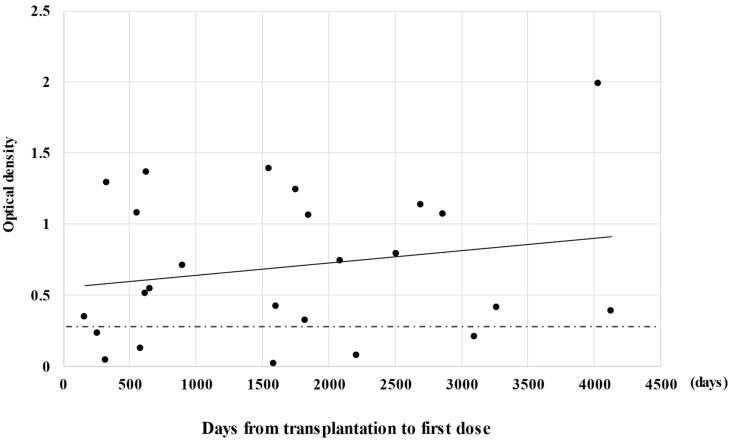
Relationship between S1 titers after the second dose and duration from transplantation to vaccination. The perforated line indicates the threshold (0.26) for seroconversion and the solid line indicates the regression line.

**Table 1 vaccines-10-00158-t001:** Patient characteristics.

Median Age at Vaccination (Range, Years)	55 (23–71)	
**Sex**	n	%
Female	11	44
Male	14	56
**Conditioning**		
MAC	9	36
RIC	16	64
**Disease**		
AML	11	44
ALL	6	24
ML	5	20
Others	3	12
**Immunosuppressants**		
Tacrolimus alone	2	8
Tacrolimus + steroid	1	4
Cyclosporine A +steroid	1	4
Steroid alone	3	12
No use	18	72
**IgG**		
≥600 mg/dL	21	84
<600 mg/dL	4	16
**Absolute lymphocyte counts**		
≥1000/μL	19	76
<1000/μL	6	24

MAC, myeloablative conditioning; RIC, reduced-intensity conditioning; AML, acute myeloid leukemia; ALL acute lymphoblastic leukemia; ML, malignant lymphoma.

**Table 2 vaccines-10-00158-t002:** Adverse events.

1st Dose	Patients, n (%)	HV, n (%)	*p* Value
Fever	1 (4)	1 (5)	1
Pain	20 (80)	12 (63)	0.308
Redness	1 (4)	0 (0)	1
Swelling	2 (8)	2 (11)	1
Headache	4 (16)	0 (0)	0.122
Fatigue	5 (20)	4 (21)	1
Chills	2 (8)	0 (0)	0.498
Muscle pain	3 (12)	1 (5)	0.622
Joint pain	2 (8)	0 (0)	0.498
Vomiting	1 (4)	0 (0)	1
Diarrhea	1 (4)	1 (5)	1
**2nd dose**			
Fever	1 (4)	2 (11)	0.181
Pain	17 (68)	13 (68)	1
Redness	2 (8)	1 (5)	1
Swelling	6 (24)	2 (11)	0.433
Headache	7 (28)	1 (5)	0.111
Fatigue	13 (52)	6 (32)	0.227
Chills	2 (8)	1 (5)	1
Muscle pain	3 (12)	1 (5)	0.622
Joint pain	1 (4)	0 (0)	1
Vomiting	0 (0)	0 (0)	1
Diarrhea	1 (4)	0 (0)	1
Skin rash	1 (4)	0 (0)	1

HV, healthy volunteers.

## Data Availability

The data presented in this study are available on request from the corresponding author.

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
