# Peer review of "The Safety and Immunogenicity of the BNT162b2 mRNA COVID-19 Vaccine in Japanese Patients after Allogeneic Stem Cell Transplantation"

_vaccines, 2022, doi:10.3390/vaccines10020158_

Round 1

Reviewer 1 Report

Marika Watanabe and colleagues present a high quality and well-written experimental manuscript that describes safety and immunogenicity of BNT162b2 mRNA COVID-19 vaccine in Japanese patients after allogeneic stem cell transplantation.

Authors prospectively evaluated the safety and immunogenicity of BNT162b2 mRNA COVID-19 vaccine (Pfizer BioNTech) in 25 Japanese allogeneic HSCT patients in comparison with 19 healthy volunteers. While anti-S1 antibody titers in almost all healthy volunteers after the second dose were higher than the cut-off value reported previously, levels in HSCT patients after the second dose were diverse.

Authors observed that 19 patients (76%) got seroconversion of anti-S1 IgG. Median optical density of antibody levels in HSCT patients with low IgG levels (< 600 mg/dL), steroid treatment, or low lymphocytes (< 1000/uL) was significantly lower than that in the other HSCT patients. There were no serious adverse events (> Grade 3), no new development or exacerbation of graft-versus-host disease after vaccination.

Authors claim that this study is the first prospective observational study on the safety and immunogenicity of BNT162b2 in Japanese patients who received allogeneic HSCT. Most HSCT patients obtained certain immunogenicity without severe adverse events, whereas patients taking steroids or with low lymphocyte counts did not yield sufficient antibody titers.

Finally, authors conclude that BNT162b2 mRNA vaccine is safe and effective in Japanese allogeneic HSCT patients.

Overall, the manuscript is highly valuable for the scientific community and should be accepted for publication.

======================

Other comments to authors:

1) Please check for typos throughout the manuscript.

2) Authors are kindly encouraged to cite the following article that describes the use of certain immunotherapies against COVID-19.
DOI: 10.3390/biomedicines9010059

Author Response

Thank you very much for your comments and suggestion about our manuscript.

1) Please check for typos throughout the manuscript.

Response 1

Thank you very much for this comment. This manuscript has been re-reviewed by an English editing service.

2) Authors are kindly encouraged to cite the following article that describes the use of certain immunotherapies against COVID-19.
DOI: 10.3390/biomedicines9010059

Response 2

Thank you for your suggestion. We have cited the article you kindly recommended in the Introduction. L.44-46

“Prevention of COVID-19 is therefore critically important in these patients while some treatments, including immunotherapy, are considered to be promising strategies against COVID-19 [2] [3].

Reviewer 2 Report

This is an important study that prospectively evaluated the safety and immunogenecity of BNT162b vaccine in Japanese allo-HSCT patients. There are some points that I am concerned about, so I would like to mention them below. I hope that my comments will help to improve this manuscript.

Major points

・L188-190

Authors should cite papers with evidence to support this sentence. (e.g., Calcineurin inhibitors do not affect vaccine efficacy in other vaccines)

・L188-190

Since this study focused on patients in the late post-transplant period, it seems inappropriate to conclude the impact on GVHD based on this number of cases. If the results differ from those of previous studies, the cause of the difference should be discussed. It is also unclear why BNT162b is considered to be safe without the evaluation of cytopenia.

・L166-169

The reasons why the results differ from previous studies should be considered.

Minor points

・If adverse reactions are to be evaluated, it should be stated how long the observation period was set after vaccination.

Author Response

Thank you very much for your time and important comments.

Major points

・L188-190 (L.185-186?)

Authors should cite papers with evidence to support this sentence. (e.g., Calcineurin inhibitors do not affect vaccine efficacy in other vaccines)

Response 1

Thank you very much for this comment.  The sentence (“Calcineurin inhibitors are known to suppress T lymphocytes mainly, because of which there might be no differences of anti-S1 IgG titers with or without calcineurin inhibitors.”) simply indicates our speculation. Generally, calcineurin inhibitors affect vaccine efficacy.  Our original sentence may have misled readers, so we have deleted the sentence, and added the following sentence instead. (L. 185-186)

“The actual impact of calcineurin inhibitors on vaccination response is unclear.”

・L188-190

Since this study focused on patients in the late post-transplant period, it seems inappropriate to conclude the impact on GVHD based on this number of cases. If the results differ from those of previous studies, the cause of the difference should be discussed. It is also unclear why BNT162b is considered to be safe without the evaluation of cytopenia.

Response 2

Thank you very much for this important comment. The previous report found that exacerbation of GVHD was observed within the first week after the first and the second injection. In our study, there were no patients who experienced development or exacerbation of GVHD in this period. Indeed, our sample size might have been too small and the observational period too short to evaluate late-onset adverse events, including GVHD. We have added this in the limitations part and changed the last sentence in the Conclusion, as below.

“Fourth, the observation period might have been too short to evaluate late-onset adverse events, including GVHD.” (L.208-209)

“Further investigation to confirm our findings is needed, and long-term monitoring of antibody levels and late-onset adverse events, including GHVD, is warranted.” (L.217-219)

In terms of cytopenia, we agree with you. When cytopenia develops, infection and bleeding events are major clinical problems. In our cohort, no patients experienced infection or bleeding events soon after vaccination.

We have added following sentence.

“; nevertheless, no patients experienced infection or bleeding events soon after vaccination.” (L.191-192)

・L166-169

The reasons why the results differ from previous studies should be considered.

Thank you very much for this comment. The reason for the different results derived from the small number of patients who underwent allogeneic stem cell transplantation within 1 year before vaccination. We have added the following sentence. (L. 174-175)

“, possibly due to the small number of patients who underwent allogeneic stem cell transplantation within 1 year before vaccination.”

Minor points

・If adverse reactions are to be evaluated, it should be stated how long the observation period was set after vaccination.

Response

We apologize for this omission. We added the following sentence.

“Vaccine-related adverse events were evaluated for 28 days after vaccination by… “(L.73)

Reviewer 3 Report

This is a thorough summary of an experience with the BNT162b1 mRNA COVID-19 vaccine in Japanese patients who have undergone allografting. 

Comments:

  1. It is never made clear how the patients were selected.  Were these consecutive patients or just 25 who were identified who received this vaccine.
  2. Likewise with the controls, there seems to be an age discrepancy in the mean age of subjects. 
  3. In Figure 3, it would be helpful to show a regression line.

Author Response

Thank you very much for your kind comments. 

Comments:

1. It is never made clear how the patients were selected.  Were these consecutive patients or just 25 who were identified who received this vaccine.

Response 1

Thank you for this comment. The patients were recruited consecutively. We have added text to clarify this.

“Consecutive patients who had undergone HSCT…” (L. 65)

2. Likewise with the controls, there seems to be an age discrepancy in the mean age of subjects. 

Response 2

Thank you for this comment. Unfortunately, we could not collect young healthy volunteers.

3. In Figure 3, it would be helpful to show a regression line.

Response 3

Thank you for your recommendation. We add the regression line as a solid line in Figure 3.

Round 2

Reviewer 2 Report

Thank you for accepting my suggestions. I have no other comments on your manuscript. These data can be very important and useful in our daily practice.